# Does the Change of Weather Influence Disease Activity in Rheumatoid Arthritis Patients: Patients’ Self-Assessment via WebApp

**DOI:** 10.3390/jcm13175336

**Published:** 2024-09-09

**Authors:** Martin Poller, Martin M. P. Schulz, Hendrik Schulze-Koops, Diego Kyburz, Johannes von Kempis, Ruediger B. Mueller

**Affiliations:** 1Department of Rheumatology, University Hospital Basel, University of Basel, 4031 Basel, Switzerland; diego.kyburz@usb.ch; 2AbbVie AG, 6330 Cham, Switzerland; martin.schulz@abbvie.com; 3Division of Rheumatology and Clinical Immunology, Department of Internal Medicine IV, Ludwig-Maximilians-University Munich, 80336 Munich, Germany; hendrik.schulze-koops@med.uni-muenchen.de; 4Division of Rheumatology and Immunology, Department of Internal Medicine, Kantonsspital St. Gallen, 9000 St. Gallen, Switzerland; johannes.vonkempis@kssg.ch; 5Rheumazentrum Ostschweiz, 9000 St. Gallen, Switzerland

**Keywords:** rheumatoid arthritis, weather, app

## Abstract

**Objectives:** The aim was to evaluate the influence of weather parameters on disease activity assessed by Routine Assessment of Patient Index Data (RAPID) scores via a Web-based smartphone application (WebApp). **Methods:** Correlation of changes of temperature (change of temperature, °C) and air pressure (change of air pressure, hPa) two days prior to and weekly self-assessment of disease activity by RAPID-3 scores over three months. To define background noise and quadrants of weather changes, we defined a central quadrant ± 2 hPa and ± 2° C, called E1. Based on this inner square, four quadrants were defined: A1 = sector left side above with increasing temperature and air pressure (improving weather); B1 = sector right side above; C1 = decreasing temperature and air pressure sector right side down (worsening weather); and D1 = sector left side down. Alterations of RAPID-3 scores analyzed changes in disease activity compared to RAPID-3 scores detected one week in advance. **Results:** Eighty patients were included in the analysis (median RA duration, 4.5 years; age, 57 years; 59% female). Median disease activity was 2.8 as assessed by DAS 28. In total, 210 time points were analyzed for quadrant A1, 164 for quadrant B1, 160 for quadrant C1, 196 for quadrant D1, and 145 for the inner square E1 were found during follow-up. The middle square E1 was balanced between increasing or decreasing values for RAPID scores. The odds for increasing RAPID scores were 1.33 (95% confidence interval CI: 1.0–1.78) for patients with ameliorating weather conditions which improve or alleviate unfavorable or adverse conditions (A1) compared to 0.98 (CI: 0.67–1.45) for worsening weather (C1) as defined by temperature and air pressure. **Conclusions:** On average, more patients developed a slight increase of disease activity if they were in the quadrant with increasing temperature and air pressure (improving weather). Thus, no correlation between the worsening of the weather and changing RAPID-3 scores was found.

## 1. Introduction

Rheumatoid arthritis is a chronic inflammatory disease characterized by joint inflammation and, subsequently, joint destruction leading to disability [1]. Many therapeutics have demonstrated efficacy by changing the course of the disease [2]. Besides genetic and immunological factors, environmental circumstances influence the development and course of the disease [3].

Such factors are cigarette smoking [4] alcohol consumption, silica dust, solvents, air pollution, ultraviolet light, reproductive and hormonal factors, such as breastfeeding [5], smoking [6], dietary factors [7], and periodontitis [3].

Claiming that weather changes, which are defined in long-term shifts in temperatures and weather patterns, may influence the course of the disease is a hypothesis that patients frequently communicate. Patberg et al. [8] from the Netherlands reviewed in 2004 the classic opinion, “cold and wet is bad, warm and dry is good for RA patients”. They found that this thesis is rather true for humidity than for temperature [9]. On the other hand, Azzouzi et al. from Morocco found no correlation between seasonal weather changes and disease activity of RA [10]. The patient numbers analyzed in these studies were small, and the outcome parameters were rarely objective. Thus, regional and ethnic differences may have influenced the published results.

In the COmPASS study [11], we analyzed the changes in RAPID-3 self-assessment via smartphones over three months weekly. The data showed a good correlation between DAS 28 scores [11,12] assessed by the treating physician and patient self-assessed RAPID-3 and 4 scores [13]. We had the opportunity to correlate the regional weather condition with the individual assessment of RAPID-3 scores every week for these patients. This enabled us to analyze whether changes may correlate with patient-related outcomes.

This study tested the hypothesis that worsening of the weather may correlate with an increase in disease activity, as frequently communicated by our patients.

## 2. Materials and Methods

### 2.1. Study Population and Design

This study is a post hoc analysis of data from a prospective, multi-center study involving 80 patients diagnosed with rheumatoid arthritis (RA) according to the 2010 criteria. [14] The inclusion criteria required patients to start a new course of conventional or subcutaneous DMARDs or both, per the Swiss label. The study followed Good Clinical Practice guidelines and the Declaration of Helsinki. Ethics committee approval was obtained at each study center, and all patients provided written informed consent (EKSG 12/135, Votum date is 26 September 2012).

### 2.2. Data Collection

Disease Activity Assessment: Patients were provided with smartphones to record their disease activity using RAPID-3 weekly over three months. RAPID-3 scores focus on physical function, pain, and patient global assessment.

Weather Data Collection: Weather data, specifically temperature and air pressure, were collected from MeteoSchweiz at the time of each patient’s disease activity assessment. Additionally, weather data were collected 48 h before each data entry. Changes in temperature and air pressure were calculated based on these measurements.

### 2.3. Definition of Weather Change Quadrants

To assess the correlation between weather changes and disease activity, changes in temperature and air pressure were categorized into five quadrants:E1 (Central Quadrant): ± 2 hPa and ± 2 °C, representing minimal change;A1 (Improving Weather): Increase in temperature and decreasing of air pressure;B1: Increase in temperature and decrease in air pressure;C1 (Worsening Weather): Decrease in temperature and increasing air pressure;D1: Decrease in temperature and increase in air pressure.

The central quadrant was expanded defining E2 and E3 by increasing the central square to ± 4 hPa and ± 4 °C (E2) and ± 6 hPa and ± 6 °C (E3), subsequently defining new outer Auadrants A2–D2 and A3–D3, respectively.

### 2.4. Statistical Analysis

Demographic and Baseline Characteristics: Demographic data and baseline disease activity parameters were analyzed for all patients and per group according the five quadrants (A1–E1).

Correlation Analysis: Odds ratios were employed to analyze the correlation between changes in weather (temperature and air pressure) and changes in RAPID-3 scores. In detail, the odds ratios for the number of patients per quadrant with increasing RAPID-3 scores vs patient with decreasing RAPID-3 scores were calculated. In parallel, the relative increase or decrease of RAPID-3 scores were analyzed per group and demonstrated as mean change. Chi-square tests were employed to analyze the statistical differences comparing the groups. Compare the mean changes in RAPID-3 scores across different quadrants. T-tests and ANOVA analyses were employed to determine if the differences in mean changes of RAPID-3 scores. Mixed-effects models specify the patient as a random effect and weather changes as fixed effects. In addition, 95% confidence intervals were calculated employing bootstrapping.

We used statistical software (e.g., R, SAS, or Stata) to fit the mixed-effects model or GEE.

Sensitivity Analysis: To determine if temperature or air pressure predominantly influenced the results, the axes were rotated by 45 degrees, creating new quadrants (A’–D’), and the analysis was repeated as mentioned above.

### 2.5. Statistical Tests and Metrics

A commonly used threshold for significance is *p* < 0.05.

## 3. Results

### 3.1. Patient Demographics

Eighty patients were included in this analysis. Patients suffered from an established RA with an average disease duration of 4.5 years. Disease activity was low, as depicted by a median DAS 28 of 2.4 and CDAI of 7.6. In total, 31.3% of the patients were treated with biologic DMARDs (Table 1).

### 3.2. Weather Data

For 24 data entries, air pressure and/or temperature change data were missing. Weather changes were depicted as relative changes in temperature and humidity (Figure 1).

In detail, 210 temperature and air pressure changes defined quadrant A1, 164 defined quadrant B1, 160 defined quadrant C1, 196 defined quadrant D1, and 145 defined the inner quadrate E1 during the follow-up at three months.

### 3.3. Demographic Data Depends on the Weather Change

Baseline demographics per patient and the respective data entry per patient (multiple data entries per patient and in different categories) were analyzed depending on the definition of the weather change defining groups A1 to E1. No differences were found for baseline demographics and parameters of disease activity depending on the defined groups (Table 1). No differences were found when the definition of the quadrants was changed to A2 to E2 or A3 to E3. 

### 3.4. Change of RAPID-3 Depending on the Change in Weather

After defining the change of air pressure and temperature into four groups, we analyzed the change of self-assessed RAPID-3 scores per patient and event and correlated it to the defined weather changes in groups A to E. On average, more patients developed a slight increase in disease activity if they were in the quadrant with increasing temperature and air pressure (improving weather conditions). This was not found for the data entries into the other quadrants B–E or the definition of the size of the inner quadrate E 1–3 (Figure 2A,C,E). In detail, 105 (57.1%), 77 (57.9%), and 52 (63.6%) patients of group A1 to A3 (ameliorating weather) developed an increase in RAPID-3 by on average 2.72, 3.06, and 2.68 points, respectively. In comparison, 51 (49.5%), 51 (49.5%), and 31 (43.1%) patients of group C1 to C3 (worsening of the weather) developed increasing values of RAPID by 2.01, 2.01, and 1.75 points, respectively (Figure 2B,D,E).

To analyze whether the temperature or the air pressure predominantly defined the weather change, the groups were redefined as A’ to D’ with the same inner quadrate E1- E3 by rotating the axes by 45° to the right (Figure 3A). Again, the demographic data for all groups were defined and revealed no differences for either group. When the absolute changes (Figure 3C) and frequencies (Figure 3B) of RAPID changes were analyzed, no differences were found for the respected subgroups.

## 4. Discussion

In summary, we showed that there was a trend toward increasing scores of RAPID-3 after an increase in temperature and air pressure, which could be defined as an amelioration of the weather. As higher RAPID-3 is equivalent to higher disease activity, we think our data falsified the RA aggravation hypothesis after worsening our test system’s weather.

However, the hypothesis remains that individuals can feel the individual worsening depending on a weather deterioration. This question cannot be addressed in our test system. Still, the perception of improving or worsening the weather is very individual. Aikman et al. [15] suggested that decreasing temperature and increasing humidity are associated with increased pain and rigidity. Strusberg et al. [16] found a correlation between lower temperature or air pressure and increasing pain. They discussed these findings to be influenced by a psychological susceptibility to weather changes described by Redelmeier et al. [17]. A direct influence driven by increasing air pressure or temperature could not be found in our analysis (Figure 3). Similarily, Staalesen Strumseet et al. found a beneficial effect of climate change from Norway to a Mediterranean setting in patients suffering from Spondyloarthritis [18].

The definition of a weather change is complex. It is commonly acknowledged that we, as individuals, understand the nature of a weather change intuitively. However, to put this intuitive perception of weather change into continuous variables such as changes in temperature and/or air pressure may be difficult. This translation of measurable variables may not correlate with the individual understanding of a weather change. We used a definition of an increasing temperature, which leads to raising of the air and, subsequently, lowering the air pressure on the ground during this process.

However, we might have neglected essential facts about weather and environmental conditions leading us to falsification of the hypothesis that worsening of weather may correlate with individual aggravation of disease activity. Similarly, Smedslund et al. [19] discussed that weather sensitivity may be detected with a broad spectrum ranging from no to high weather sensitivity. Similar to our data, about 57% of the patients developed increasing RAPID scores in parallel to improving weather conditions (Odds 1.33). In parallel to Smedslund et al. [19], we agree that there may be a variation in weather sensitivity. For example, Xie et al. [20] found an association between healthcare and local rainfalls.

The correlation of weather change over the last two days with a change in disease activity over the last week may appear to be counter-intuitive. Disease activity usually changes slowly. However, within a week, the weather may change back and forth more than once. Therefore, correlating the actual weather condition with the weather detected in the last data entry seems inappropriate to describe the reaction toward a current weather change. Subsequently, a change in disease activity was correlated to a detection one week before.

Importantly, patients in our cohort were frequently in LDA and under continuous treatment with biologic and/or synthetic DMARDs. Therefore, a weather change may be minor compared to the potential change of disease activity in a clinical trial and a newly invented drug. This bias, however, cannot be ruled out ethically.

### 4.1. Advantages

To our knowledge, this is the first analysis with weekly data on weather conditions and RAPID scores. This enabled us to objectively analyze parameters indicative of the weather and disease activity.

The patients were unaware of this post hoc applied question on the data. Therefore, no expectation bias of the analyzing physicians and/or patients could have influenced the data.

### 4.2. Limitations

A potential bias may be that the disease activity was compared to the last assessment of disease activity one week earlier, whereas the weather change was analyzed over the preceding two days. A potential weather change influencing the first measurement of RAPID-3 and leading to an artificial change of disease activity as assessed by RAPID-3 cannot be ruled out.

A second limitation may be that the location for the weather analysis was based on the institution. Therefore, for patients living far away from the respective institution, unique local weather phenomena can occur. Secondly, the patients may have been traveling to a different region at the time of the assessment. The correct location data of the smartphones when the assessment was conducted could not be obtained for data protection reasons.

Furthermore, a potential limitation is the time frame from November 2012 to March 2014 when the study was conducted. As weather changes are variable between years, seasons, and regions, it is difficult to compare ours with other studies.

An indirect effect of a weather change on the individual psychological condition, leading to a different judgment on individual RAPID questions, cannot be ruled out.

It is difficult to define a change in the weather. There is no reliable definition of what we would commonly consider good or bad weather. Individual variations may further influence this judgment. Whether such an analysis may be transferred to other climatic conditions with more rain, less humidity, or higher temperatures remains open.

## Figures and Tables

**Figure 1 jcm-13-05336-f001:**
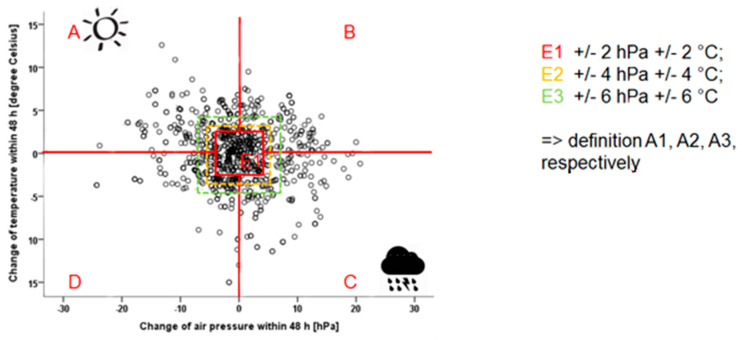
Change of the weather conditions was analyzed by assessing the change of air pressure in hPa (x axis) and change of temperature in ° Celsius (y axis) between two measurements. Middle Square (Red−E1): The smallest square in the center, shown in red, represents “no significant change” in weather conditions. This square includes changes within a range of ±2 hPa for air pressure and ±2 °C for temperature. Quadrants (A, B, C, D): The rest of the graph outside the red square is divided into four quadrants (labeled A, B, C, and D), each corresponding to different combinations of increases or decreases in air pressure and temperature. Expanded Squares: The red middle square was then expanded to form larger squares, each representing a wider range of weather changes. Yellow Square (E2): Includes changes within ±4 hPa and ±4 °C. Green Square (E3): Includes changes within ±6 hPa and ±6 °C. As these squares grow, they form new quadrants (A2–D2 for the yellow square E2, A3–D3 for the green square E3), allowing us to analyze weather changes across broader ranges.

**Figure 2 jcm-13-05336-f002:**
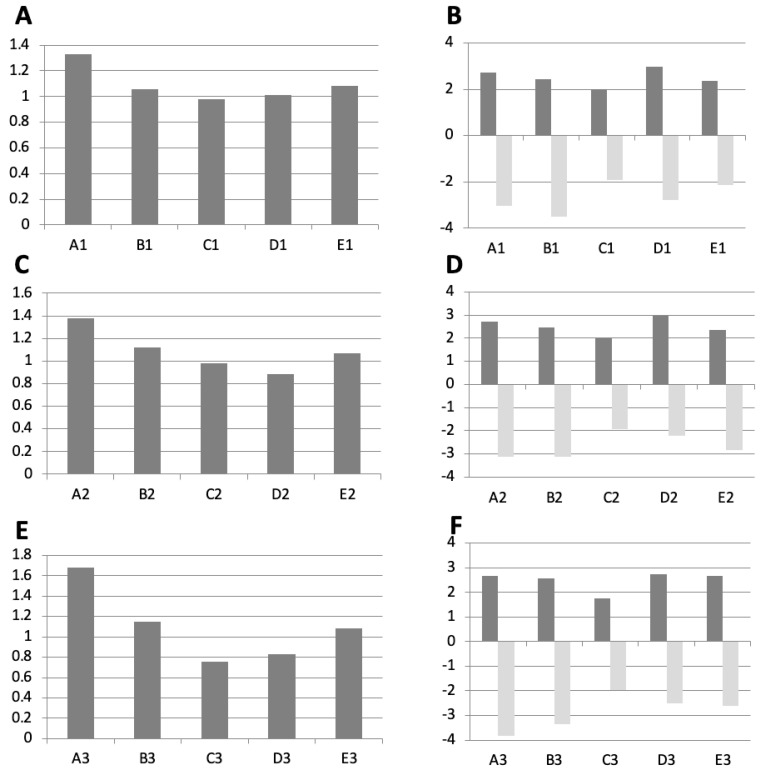
Change of RAPID-3 scores between two measurements were analyzed for all patients and visits. RAPID−3 were correlated with changes in weather, defined by changes in air pressure and temperature per group (**A**–**E**), as defined in Figure 1. Odds Ratios (panels (**A**,**C**,**E**)): the ratio of the number of episodes where the RAPID-3 scores increased as compared to those where they decreased is demonstrated for each quadrant and square size. This shows how likely RAPID-3 score increased or decreased with changes in weather conditions. Mean increases and decreases of RAPID-3 (panels (**B**,**D**,**F**)): This shows the average amount by which the RAPID-3 scores either increased or decreased in each quadrant and for each square size. It provides a more detailed view of the magnitude of change in disease activity in relation to the weather.

**Figure 3 jcm-13-05336-f003:**
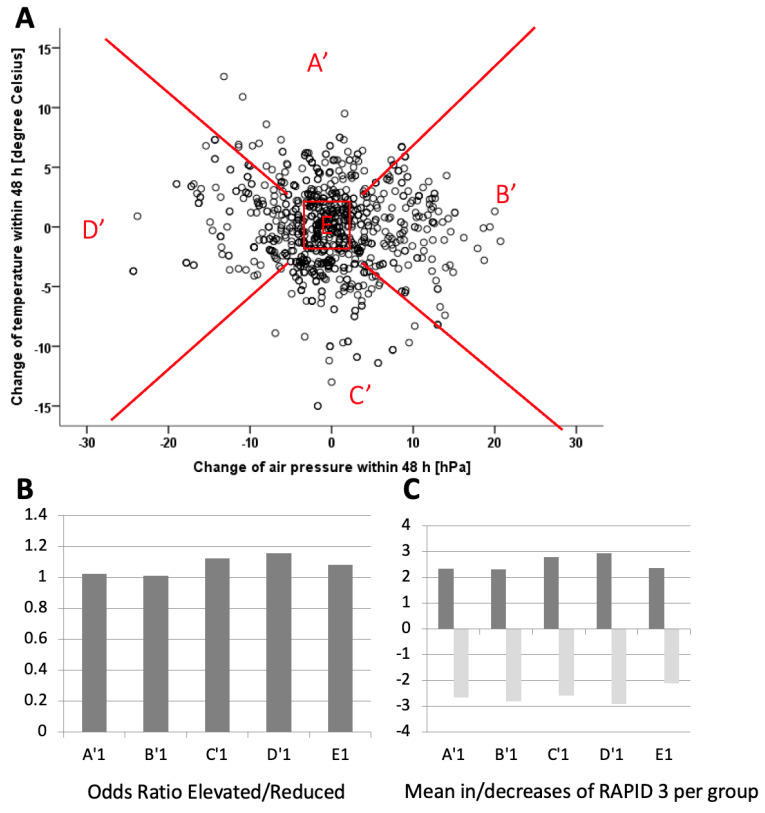
(**A**) The study analyzed changes in weather conditions between two patient visits by measuring variations in air pressure (hPa) on the x-axis and temperature (°C) on the y-axis. A central reference square, labeled “E” and colored red, was established to represent conditions with minimal or no change, defined as a shift of ±2 hPa in air pressure and ±2 °C in temperature (upper panel). This square served as the baseline for comparison. Surrounding the central square, four rotated quadrants (**A**) were defined: A1’, B1’, C1’, and D1’. Each quadrant represents a distinct combination of changes in air pressure and temperature: Quadrant A’: Increasing temperature; Quadrant B’: Increasing air pressure; Quadrant C’: Decreasing temperature; Quadrant D’: Decreasing air pressure. In addition to analyzing weather changes, the study also examined changes in RAPID-3 scores—a measure of disease activity—between the same two visits for all patients. The analysis focused on how these score changes corresponded to the weather changes categorized by the four quadrants (A1’, B1’, C1’, D1’) and the central square “E.” (**B**) Odds Ratios (lower left panel): the ratio of the number of episodes where the RAPID-3 scores increased compared as compared to those where they decreased is demonstrated for each quadrant and square size. This shows how likely RAPID-3 score increased or decreased with changes in weather conditions (A’–D’, and E1). (**C**) Mean increases and decreases of RAPID-3 (lower right panel): This shows the average change of RAPID-3 scores either increased or decreased in each quadrant (A’–D’, and E1). It provides a more detailed view of the magnitude of change in disease activity in relation to the weather (Table 2).

**Table 1 jcm-13-05336-t001:** Patient characteristics.

	At Baseline (*n* = 80)
Female	58.8%
Age, median, years (IQR)	57.3 (44.9–65.9)
RA duration since diagnosis, median, years	4.5 (1.1–9.5)
DMARD use (%)	82.5%
Biologic DMARD use (%)	31.3%
DAS 28, median (IQR)	2.8 (1.9–3.5)
CDAI, median (IQR)	7.6 (4.3–16.5)
SDAI, median (IQR)	10.5 (6.6–20.4)
RAPID3, median (IQR)	6.6 (3.1–11.9)
HAQ-DI, median (IQR)	0.38 (0.13–1.13)

IQR: interquartile range.

**Table 2 jcm-13-05336-t002:** Baseline demographics, depending on the change in temperature and air pressure.

	All	Group A1	Group B1	Group C1	Group D1	Group E1
Number of assessments	875	210	164	160	196	145
Female, %	62.2	59.0	58.5	63.8	64.3	66.2
Age, years (mean, SD)	55.1 (13.3)	54.5 (14.4)	54.9 (13.2)	54.4 (12.2)	56.6 (12.5)	54.6 (14.1)
RA duration since diagnosis, years (mean, SD)	7.32 (9.28)	7.3 (9.48)	7.91 (9.34)	6.21 (8.0)	8.0 (9.83)	6.97 (9.46)
DMARD use, %	91.4	90.0	87.8	93.8	93.4	92.4
Biologic DMARD use, %	35.8	39.5	35.4	36.9	36.7	28.3
DAS 28 (ESR, mean, SD)	2.68 (1.44)	2.55 (1.36)	2.85 (2.81)	2.81 (1.38)	2.61 (1.44)	2.75 (1.55)
CDAI (mean, SD)	11.2 (10.2)	10.4 (9.28)	11.1 (10.3)	12.7 (12.0)	10.9 (9.5)	11.2 (10.2)
RAPID-3 (mean, SD)	7.99 (5.56)	8.29 (5.82)	7.99 (5.37)	8.26 (5.54)	8.26 (5.54)	7.37 (5.47)
HAQ-DI (mean, SD)	0.7 (0.73)	0.72 (0.77)	0.68 (0.74)	0.68 (0.75)	0.7 (0.7)	0.71 (0.71)

## Data Availability

The original contributions presented in the study are included in the article, further inquiries can be directed to the corresponding authors.

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
