# Peer review of "Does the Change of Weather Influence Disease Activity in Rheumatoid Arthritis Patients: Patients’ Self-Assessment via WebApp"

_jcm, 2024, doi:10.3390/jcm13175336_

Round 1

Reviewer 1 Report

Comments and Suggestions for Authors

This is an interesting study investigating the relationship between weather changes and disease activity of RA patients. The result showed that higher temperature and high air pressure were related with disease activity, which is contrary to the general belief.  There are several concerns about the manuscript.

Statistical methods.

The statistical section is too short. How did you define correlation between weather changes and disease activity? That is to say, how was the significance defined?
It is expected that the data include correlated data since multiple measurements were performed in each patient. How were the correlated data treated statistically?

Table 1.

For values represented by median, please show the quatile ranges.

Figure 1.

Usually the low air pressure is related with high chance of rain. But, probability of rain seems to be associated with higher air pressure in this figure

Figure 2.

It is difficult to see the figures. They were broken.  It is difficult to understand the meaning of the figures.  The legends of the figures need to be rewritten.

Author Response

Reviewer: Statistical methods.

The statistical section is too short. How did you define the correlation between weather changes and disease activity? That is to say, how was the significance defined?

It is expected that the data include correlated data since multiple measurements were performed in each patient. How were the correlated data treated statistically?

Our answer: The respected reviewer is right. We revised the complete methods section and described the statistical methods in detail.

Reviewer: For values represented by a median, please show the quartile ranges.

Our answer:  By reporting the median along with the quartile ranges, you provide a clear and comprehensive summary of the central tendency and variability in your data. This approach is particularly useful for skewed distributions or when outliers are present, as it gives a more robust measure of central location compared to the mean. However, we prefer to show the data as mean, and SD as outliers are not a major issue for this analysis.

Reviewer: Figure 1. Usually, the low air pressure is related to a high chance of rain. But probability of rain seems to be associated with higher air pressure in this figure.

Our answer: A definition of a weather change does not exist and is difficult to make despite the fact that we think that we may feel it precisely. If the air heats up, it rises, and the air pressure on the ground decreases with improving weather conditions. The icons with a sun were rather inserted to improve the readability. However, these icons do not reflect the occurrence of rain but rather symbolize the worsening of the weather conditions. The figures were completely revised.

Reviewer: Figure 2. It is difficult to see the figures. They were broken.  It is difficult to understand the meaning of the figures.  The legends of the figures need to be rewritten.

Our answer: The document's formatting was changed to improve the figure's readability, and the legends were rewritten.

Reviewer 2 Report

Comments and Suggestions for Authors

This is a topic which rheumatologist meet nearly every day in their daily clinical practice.  Patients often complain about worsening of their rheumatology symptoms when weather is worsening (cold and wet) and patients do often feel better in warm and stable climates. Interesting for RA patients this was not found in the present study. There were only a small signal and this went in the opposite direction where “improving” weather did not cause lower perception of disease activity.

I find the tittle some misleading and would recommend re-phrasing as the main outcome is the patient perception of disease activity when using RAPID-3 as outcome measure.

Minor issue: Abstract median DAS28 is reported to be 2.8 in table 1 2.4.

Patients had in general low disease activity as assessed with DAS28 which then questions if the study was at all designed to test if climate do improve true inflammatory disease activity, however said this this study addresses perception of disease activity.

When it comes to weather sensitivity others have reported this to be a highly individual phenomenon (ref 18). So to capture changing weather phenomenon’s and impact on patients perception of disease may also be difficult. If weather is changing rapidly this may be difficult to capture as the authors also addresses.

I would recommend the authors to have a look at these to studies where impact of climate is tested for a longer period one study for AS and one for RA and address this in the discussion.

·         Staalesen Strumse YA, et al . Efficacy of rehabilitation for patients with ankylosing spondylitis: comparison of a four-week rehabilitation programme in a Mediterranean and a Norwegian setting. J Rehabil Med. 2011 May;43(6):534-42. doi: 10.2340/16501977-0804. PMID: 21491073.

·         Staalesen Strumse YA, et al . The efficacy of rehabilitation for patients with rheumatoid arthritis: comparison between a 4-week rehabilitation programme in a warm and cold climate. Scand J Rheumatol. 2009 Jan-Feb;38(1):28-37. doi: 10.1080/03009740802304549. PMID: 18728936.

Author Response

Reviewer: I find the tittle some misleading and would recommend re-phrasing as the main outcome is the patient perception of disease activity when using RAPID-3 as outcome measure.

Our answer: The word perception was used twice in the text, in lines 527 and 538. In both parts, the word perception does not refer to the RAPID-3 scores but rather to the perception of an improving weather condition. We think that the wording in these two cases is correct. When we introduced RAPID3 in the methods, we wrote "self-assessment of disease activity by RAPID-3 scores," which we consider to be correct.

Reviewer: Minor issue: Abstract median DAS28 is reported to be 2.8 in table 1 2.4.

Our answer: The typo was corrected in table 1. Thank you for pointing this out.

Reviewer: Patients had low disease activity in general as assessed with DAS28, which then raises questions about whether the study was designed to test whether climate does improve true inflammatory disease activity. However, the reviewer said this study addresses perception of disease activity.

Our answer: This is right. The study was rather designed to detect flare-ups during follow-up in patients with low disease activity. Our analysis is a post hoc analysis on the back of these data.

Reviewer: When it comes to weather sensitivity others have reported this to be a highly individual phenomenon (ref 18). So to capture changing weather phenomenon’s and impact on patients perception of disease may also be difficult. If weather is changing rapidly this may be difficult to capture as the authors also addresses.

Reviewer: I would recommend the authors to have a look at these to studies where impact of climate is tested for a longer period one study for AS and one for RA and address this in the discussion.

  • Staalesen Strumse YA, et al . Efficacy of rehabilitation for patients with ankylosing spondylitis: comparison of a four-week rehabilitation programme in a Mediterranean and a Norwegian setting. J Rehabil Med. 2011 May;43(6):534-42. doi: 10.2340/16501977-0804. PMID: 21491073.
  • Staalesen Strumse YA, et al . The efficacy of rehabilitation for patients with rheumatoid arthritis: comparison between a 4-week rehabilitation programme in a warm and cold climate. Scand J Rheumatol. 2009 Jan-Feb;38(1):28-37. doi: 10.1080/03009740802304549. PMID: 18728936

Our answer: The fact that changing the weather conditions is difficult to define is discussed in the paragraph "The definition of a weather change is complex" line 536ff. The papers of Staalesen Strumse discuss the effect of climate conditions rather than local weather changes on SpA patients. Therefore, this is a different approach. However, this important publication is now referenced in the paper.

Reviewer 3 Report

Comments and Suggestions for Authors

Dear authors,

Thank you for this paper looking at weather change & RAPID3 scores in RA. Although interesting, the low sample size coupled to the fact that weather is measured at institution level, and not at the patient level makes the results less reliable.

More bias could be that rapid3 is not the ideal instrument to measure change. It could be that elements of rapid3 are better to be analysed in this respect.

I'm also a bit surprised that the largest smartphone study on weather change and pain (cloudy with a chance of pain - dixon) is not mentioned

Author Response

Reviewer: Thank you for this paper looking at weather change & RAPID3 scores in RA. Although interesting, the low sample size coupled to the fact that weather is measured at institution level, and not at the patient level makes the results less reliable.

Our answer: The respected reviewer is right on this point. This is the closest we could get to an exact measurement. This point is discussed among limitations lines 572 - 592. However, local weather phenomena are rare and should not, overall, influence the data.

Reviewer: More bias could be that rapid3 is not the ideal instrument to measure change. It could be that elements of rapid3 are better to be analysed in this respect.

Our answer: The RAPID 3 is the outcome we had available. This is why we used it.

Reviewer: I'm also a bit surprised that the largest smartphone study on weather change and pain (cloudy with a chance of pain - dixon) is not mentioned

Our answer: The study by Reade and Dixon. 2017 Mar 24;5(3):e37. doi: 10.2196/mhealth.6496 comprises only 20 patients of which 6 dropped out and, therefore, was not referenced.

Round 2

Reviewer 1 Report

Comments and Suggestions for Authors

The queries are adequately addressed. 

Author Response

Reviewer: 

- The section on statistical methods, from lines 145 to 249, should be significantly revised. It is unnecessary to detail every possible statistical approach. Instead, focus on clarifying the specific methods used to handle your data within the manuscript. Readers are generally expected to understand basic statistical concepts like odds ratios and p-values, so an exhaustive explanation is not required. As it currently stands, this section reads more like a step-by-step guide (generated by an AI tool?), rather than the concise and precise description expected in a scientific manuscript. This portion needs revision before the article can be considered for publication.

Our answer: 

The statical section was completely revised. With the first rebut, we were running out of time because of our vacations. We apologize for this and hope that the current version suits the reviewer’s expectations. Thank you for pointing this out to us.

Reviewer: 

- All odds ratios (ORs) presented in the manuscript should be accompanied by their confidence intervals (preferably 95% CI), as this is essential for proper interpretation. Please ensure that the 95% CI is reported for each OR in both the text and figures.

Our answer: 

The 95% confidence intervals were calculated and added into the paper.

Reviewer: 

- For the figures that represent odds ratios (e.g., panels A, C, and E of Figure 2), using a Forest plot would enhance the visualization of the effect size.

Our answer: 

We think this is a good idea, and we have created the Forest blot. However, we think that readability has not increased. Therefore, we kindly ask the reviewer to keep the graph as is. The Forest blot is uploaded separately.

Reviewer: 

- In Table 1 and throughout the text, interquartile ranges (IQRs) should be provided for each median value.

Our answer: 

The IQRs are now provided

Reviewer: 

- Please correct the keyword “wheather” to “weather.”

- In lines 319-320, there is a repetition of the word “compared.” Please revise this to avoid redundancy.

Our answer: 

The misspelled weather version was corrected, and the repeated word was compared in line 320. Thank you for finding this mishap.
